# Perioperative Nutritional Management in Enhanced Recovery after Bariatric Surgery

**DOI:** 10.3390/ijerph20196899

**Published:** 2023-10-08

**Authors:** Giovanna Flore, Andrea Deledda, Michele Fosci, Mauro Lombardo, Enrico Moroni, Stefano Pintus, Fernanda Velluzzi, Giovanni Fantola

**Affiliations:** 1Obesity Unit, Department of Medical Sciences and Public Health, University of Cagliari, 09124 Cagliari, Italy; gflore@unica.it (G.F.); andredele@tiscali.it (A.D.); michele.fosci92@gmail.com (M.F.); 2Department of Human Sciences and Promotion of the Quality of Life, San Raffaele Open University, Via di Val Cannuta, 247, 00166 Rome, Italy; mauro.lombardo@uniroma5.it; 3Obesity Surgery Unit, Department of Surgery, Azienda di Rilievo Nazionale ed Alta Specializzazione G. Brotzu, 09134 Cagliari, Italy; enrico.moroni@aob.it (E.M.); stepintuss@gmail.com (S.P.); nannifantola@hotmail.it (G.F.)

**Keywords:** obesity, bariatric surgery, ERABS protocols, fasting, efficacy, metabolic comorbidities, preoperative fasting, carbohydrate loading, postoperative refeeding, perioperative nutrition, early refeeding

## Abstract

Obesity is a crucial health problem because it leads to several chronic diseases with an increased risk of mortality and it is very hard to reverse with conventional treatment including changes in lifestyle and pharmacotherapy. Bariatric surgery (BS), comprising a range of various surgical procedures that modify the digestive tract favouring weight loss, is considered the most effective medical intervention to counteract severe obesity, especially in the presence of metabolic comorbidities. The Enhanced Recovery After Bariatric Surgery (ERABS) protocols include a set of recommendations that can be applied before and after BS. The primary aim of ERABS protocols is to facilitate and expedite the recovery process while enhancing the overall effectiveness of bariatric procedures. ERABS protocols include indications about preoperative fasting as well as on how to feed the patient on the day of the intervention, and how to nourish and hydrate in the days after BS. This narrative review examines the application, the feasibility and the efficacy of ERABS protocols applied to the field of nutrition. We found that ERABS protocols, in particular not fasting the patient before the surgery, are often not correctly applied for reasons that are not evidence-based. Furthermore, we identified some gaps in the research about some practises that could be implemented in the presence of additional evidence.

## 1. Introduction

Obesity and overweight are a major health problem worldwide primarily due to the numerous related comorbidities and complications. These include cardiovascular disease, diabetes mellitus, metabolic syndrome, non-alcoholic steatohepatitis, gallbladder disease, gastroesophageal reflux, obstructive sleep apnoea, reproductive system disorders, and osteoarthritis, and can significantly reduce both the quality and the expectancy of life [1,2,3]. Also, the prevalence of many cancers (such as breast, colon, prostate, endometrium, kidney, and gallbladder) [4], as well as social and psychological problems [5,6,7], increases proportionally to the Body Mass Index (BMI) [8]. Health risk rises five-fold for people with a BMI of 25, 28-fold for a BMI of 30, and 93-fold for a BMI of 34.9 or more [9]. Clinical guidelines recommend that obesity management programs consist of an integrated multidisciplinary patient-centred approach including lifestyle interventions, pharmacologic therapy, and bariatric surgery (BS) [10].

Reversing obesity only by means of lifestyle changes is difficult [11]; insufficient weight loss and weight regain are common in clinical obesity management [12], also due to changes in the energy metabolism of people with obesity [13]. Pharmacotherapy can be used in association with lifestyle interventions, but the prescription has some limitations according to national rules [10]. BS is considered the most efficient medical intervention to counteract severe obesity, especially in the presence of metabolic alterations like diabetes or other risk factors [14].

BS is indicated for patients with a BMI > 40 kg/m^2^ or a BMI > 35 kg/m^2^ with metabolic complications [15]. However, the most recently updated guidelines from the American Society for Metabolic and Bariatric Surgery (ASMBS) and the International Federation for the Surgery of Obesity and Metabolic Disorders (IFSO) allow BS also in patients with BMI > 35 kg/m^2^ without metabolic complications [16].

There are various surgical procedures. Roux-en-Y gastric bypass (RYGB), laparoscopic sleeve gastrectomy (LSG), and one-anastomosis gastric bypass (OAGB) represent over 80% of the interventions worldwide [17]. Preoperative diet and nutritional treatment in the first hours and days after surgery may impact the outcomes of intervention, and long-term adherence to the diet is essential to preserve health [18]. An evaluation of large databases, deposited in a public archive, highlighted that patients undergoing BS may have nutritional deficiencies despite the use of supplements being recommended by guidelines [19].

Although BS is considered a safe procedure, with risks comparable to other surgical treatments like cholecystectomy [20], in the past decade the Enhanced Recovery After Surgery protocol (ERAS) has also been implemented for BS. This protocol aims to facilitate physical and mental preparation, reduce preoperative anxiety, postoperative complication rates, and length of recovery, and improve functional recovery time after surgery [21]. Indeed, the ERAS protocols refer to a multimodal stress-minimizing approach designed to reduce the rate of morbidity after major surgery [22].

Since the first ERAS guidelines, published in 2005, and particularly regarding colonic surgery, the importance of perioperative nutrition has been emphasized [23]. In the latest ERAS guidelines for colorectal surgery [24], preoperative nutritional care, including preoperative nutritional screening and nutritional status management, gained a strong recommendation with low and moderate quality of evidence. Nevertheless, preoperative carbohydrate loading was not evaluated in ERAS. ERAS items for colorectal surgery were later extended to other surgical specialities, including BS (ERABS) [25,26]. ERAS Society guidelines concerning BS (ERABS) were first published in 2016 [27] with an update in 2022 [22]. The ERABS guidelines emphasized the importance of preoperative management, focusing on preoperative weight loss and diabetes optimization (a strong recommendation with low quality of evidence), but as in the colorectal ERAS guidelines, preoperative carbohydrate loading was not mentioned. ERABS has become a well-established protocol including safe and feasible items and it was considered necessary for bariatric patients [22,28].

However, neither the ERABS protocol nor perioperative nutritional management were evaluated in the recent American Society for Metabolic Bariatric Surgery (ASMBS) guidelines [16]. The ASMBS published a national quality improvement project titled “Employing Enhanced Recovery Goals in Bariatric Surgery (ENERGY)” using the Metabolic and Bariatric Surgery Accreditation and Quality Improvement Program. However, perioperative nutritional management was not well defined [29]. On the other hand, the guidelines from the Italian Society of Obesity Surgery and Metabolic Disease recently recommended the use of ERABS items in order to improve postoperative outcomes (a strong recommendation with moderate quality of evidence). Furthermore, these guidelines also recommended perioperative nutritional management, particularly addressing vitamin D deficiency in the perioperative phase (a strong recommendation with moderate quality of evidence) [30]. Perioperative nutritional care represents an important specific component of ERABS protocol, which comprises in total 20 items [31].

The nutritional guidelines can be summarised as follows:A minimum of 6 h fasting for solids and 2 h for clear fluids before anaesthesia induction is advised under standard circumstances;Consuming iso-osmolar carbohydrate drinks (CHO) 2–3 h before anaesthesia may mitigate postoperative insulin resistance and help in retaining lean body mass, although evidence is inconclusive [32,33].

The first fasting rule is considered essential, while the benefits of preoperative carbohydrate consumption are not clearly established [22].

For postoperative nutrition, clear liquid intake can start two hours after surgery, followed by more nourishing fluids. Dietary consultations will offer texture-specific guidance based on the type of surgery. Patients should eat slowly, chew thoroughly, and avoid beverages during meals. Carbonated drinks and alcohol are not allowed (Figure 1).

Moreover, international guidelines for BS state in general terms that perioperative diet and postoperative meal progression should be under guidance of a trained nutritionist [34] and patients should have access to a comprehensive nutrition and dietetic assessment [22].

The objective of the present review is to analyse the importance and effectiveness of perioperative nutritional care within the ERABS protocol. This review aims to assess the usefulness and necessity of implementing such care and to explore the potential benefits it can provide. Additionally, the review aims to identify strategies for the effective implementation of perioperative nutritional care within the ERABS protocol. We also highlight some gaps in the research that could be added to the ERABS protocol after appropriate investigation, in particular concerning the use of supplements or nutraceuticals that may allow a faster recovery.

## 2. Methods

This is a narrative review of the literature based on articles and guidelines published in English. The search was conducted on the PubMed, Web of Science, and Cochrane databases, based on their relevance to BS and preoperative and postoperative fasting, without restriction of time. The literature search was performed using the terms “ERABS”, “bariatric surgery guidelines”, “nutritional treatment in bariatric surgery”, “preoperative nutrition”, “carbohydrate loading”, “preoperative fasting”, “postoperative refeeding”, “perioperative nutrition”, and “early refeeding”. A comprehensive selection of publications was obtained, including observational studies, randomized controlled trials (RCTs), systematic and non-systematic reviews, and thematic reviews, as well as practice and management guidelines. The exclusion criteria for the review involved excluding non-English articles to maintain language consistency. Duplicate publications were also excluded to avoid repetition, and publication types such as editorials and conference abstracts were excluded. Additionally, articles lacking sufficient data or those that did not provide relevant information on perioperative care within the ERABS protocol were excluded. Through a rigorous screening process involving the evaluation of titles, abstracts, and full texts, 180 articles were deemed relevant and included in the review. Two reviewers (A.D. and G.F.) examined the full text of the selected articles and independently judged the appropriateness of inclusion. In cases of disagreement, a third reviewer (F.V.) was invited to participate in the review procedure. Studies that were irrelevant to the research objectives, had poor methodology, inadequate sample sizes, or were duplicate publications were excluded. Out of the 180 articles initially selected for the review, 107 articles underwent a comprehensive core review process. These articles were carefully evaluated to extract relevant information and insights regarding perioperative care in the context of the ERABS protocol. Among the core review articles, 20 specifically addressed different perioperative phases, providing valuable insights into preoperative, intraoperative, and postoperative nutritional care practices within the ERABS protocol (Figure 2).

## 3. Preoperative Fasting (PF)

One of the aims of ERABS is to eliminate the habitual dogma of fasting the patient. Indeed, PF is not long suggested by ESPEN clinical guidelines, and this can be considered a part of ERAS [35]. This is consistent with the latest American Anesthesiologists Practice Guidelines for preoperative fasting, released in 2023 [36]. However, a period of up to 12 h of PF is still considered fundamental by many anaesthesiologists and surgeons. The explanation for this prescription is to increase gastric pH and reduce stomach content, decreasing nausea and emesis and potential need for aspiration [37]. Although this traditional practice was justified when anaesthetic techniques were undeveloped [38], nowadays it makes no sense, but a resistance to changes in guidelines and clinical practices was observed [39].

An analysis of the PF practices commonly used by anaesthesiologists shows that many of them are resistant to abandoning outdated procedures like forbidding fluid intake before surgery [40]. More specifically, Merchant et al. reported that a substantial proportion of anaesthesiologists responding to an internet survey in three regions (Canada, Europe, Australia/New Zealand) impose strict requirements for both solid and liquid fasting after midnight before surgery. Issues related to variable operating room hours, although it seems to occur rarely, and safety concerns linked to the implementation of clear guidelines for fluid intake were the most cited reasons for maintaining these outdated practices [40].

The reason for applying PF was mainly the risk of aspiration, which occurs in surgical patients causing transient pneumonia that in some cases can progress to severe acute clinical lung damage or acute respiratory distress syndrome [41]. However, in a study comprising 10,015 paediatric anaesthesia procedures, aspiration occurred in three cases (0.03%). No cases required cancellation of the surgical procedure, intensive care, or ventilation support, and no deaths attributable to aspiration were reported [42].

The risk of aspiration has been reviewed by many authorities, particularly in paediatric anaesthesiology. Beach et al., representing the Paediatric Sedation Research Consortium (PSRC), analysed the PSRC database and reported an aspiration rate of eight out of 82,546 fasting patients and of two out of 25,401 non-fasting patients. In diabetic patients, particularly those who have had the condition for over 10 years, a delay in gastric emptying has been observed, though conclusive data are lacking. The authors of the review concluded that studies on fasting gastric content and volume in patients with diabetes mellitus are limited, often do not include blinding of evaluators, and offer conflicting evidence; consequently, the true risk of aspiration in patients with diabetes mellitus on an empty stomach is unknown and may have been overestimated in the past [43]. Considering the complex interplay of patient intervention and anaesthetic factors for aspiration, a nil per os guideline measure is unlikely to be appropriate for all. Thus, the review suggests an individualized anaesthetic plan through a careful medical examination [44].

Surgery can be considered a traumatic condition for the human body leading to alterations in inflammatory cytokines, immune system responses, and insulin resistance, thus contributing negatively to post-operative recovery. The immunological mechanisms linked to trauma and major surgery infections involve hyperglycaemia and short-term hyperinsulinemia. Both of these conditions are closely correlated with infection-related complications and mortality. Overnight or prolonged fasting before surgery worsens the patient’s prognosis and may even induce several adverse effects such as vomiting, hunger, anxiety, and more severe symptoms.

Several studies have investigated the effect of PF in elderly populations, considered as a fragile category. A prolonged preoperative fasting time led to unfavourable outcomes after gastrointestinal surgery. Therefore, reducing the preoperative fasting time is likely to accelerate postoperative recovery in these patients [45]. However, elderly patients should not be automatically classified as high-risk per se. Instead, severe comorbidities and other surgical risk factors are more indicative of a frail condition. The focus should be on the development of a customized protocol based on the differences between surgery and the patient’s comorbidities rather than specifically on age [46].

The latest guidelines claim that there is no evidence that a shortened fast of 2 to 3 h, which includes oral fluids, increases the risk of aspiration, regurgitation, or related morbidity compared with total abstention from food and liquids from midnight [47,48,49]. Moreover, water restriction seems to be an out-dated practice considering the significant gastric volume reduction in patient drinking preoperatively.

Furthermore, PF can increase the risk of postoperative complications, like length of stay (LOS), worsening insulin resistance, thirst, hunger, malaise, weakness, and intestinal discomfort (nausea and vomiting) [50]. However, it can be reasonable in special circumstances, like gastroesophageal reflux (GERD), delayed gastric emptying, and patients “at special risk” [51].

## 4. Preoperative Carbohydrates Load (PreCL)

According to guidelines from ERAS Society, there is no sufficient evidence to recommend a carbohydrate loading before surgical procedures [22].

Apparently, PreCL does not influence nausea after a GBP procedure compared to administration of protein or water. Karlsson et al., in a randomized trial, examined the effect of three different preoperative nutrition modalities on the short-term outcome of problems such as nausea and pain, critical factors for early patient mobilization and early discharge. In this study, neither carbohydrates, nor fat and protein administration, nor only oral volume administration seemed to influence any variable measured in the postoperative setting. The results support the idea that patients undergoing surgery with short operative times and within an established ERAS program do not benefit from extra preoperative feeding, excluding at the same time any risk connected. Thus, even in absence of benefits, PreCL does not raise risks [52].

Surgical stress initiates a multifaceted response encompassing sympathetic nervous system activation, catabolic hormone release, and local cytokine reactions to tissue damage. This response generally correlates with the severity of surgical trauma. The endocrine aspect involves the activation of the hypothalamic–pituitary–adrenal axis, leading to increased cortisol, vasopressin, and pancreatic glucagon secretion. This contributes to elevated peripheral insulin resistance and muscle catabolism. A connection has been found between peripheral insulin resistance and the degree of muscle catabolic response [53].

Current surgical protocols involve interventions intended to reduce surgical stress and the degree of postoperative insulin resistance. These interventions include preoperative administration of carbohydrates, either orally or intravenously, up to two hours before surgery in contrast to traditional PF [54]. Gastric emptying studies have shown that ingestion up to 400 mL at least two hours before undergoing opioid-containing analgesia provokes a residual gastric volume equivalent to that of overnight fasting. Preoperative carbohydrate treatment stimulates endogenous insulin release, switching the metabolic state of overnight fasting and decreasing the extent of peripheral insulin resistance, improving the surgical stress response [55].

The RCT of Suh et al. [56] demonstrates that beverages with preoperative carbohydrate load can be administered in patients undergoing bariatric surgery without significant postoperative risks. More specifically, this study did not find any significant differences in LOS when the intervention group (with drink) was compared to a control group that followed a standard protocol (without drink). Moreover, none of the patients needed aspiration, and diabetic patients did not experience increased postoperative complications or alterations in postoperative glycaemic control. This finding is in line with other studies [26] that did not observe a heightened risk of aspiration, regurgitation, or related morbidity with short fasts (2–3 h) compared to fasting after midnight [56].

In a 2014 meta-analysis involving 1685 elective surgery patients, preoperative carbohydrate treatment led to a one-day reduction in hospital length of stay (LOS) for those undergoing major abdominal surgery. The improvement in recovery was not due to a decrease in complication rates, which remained consistent between the carbohydrate-treated group and the control group. The authors attributed the enhanced recovery to the metabolic benefits of the carbohydrate treatment, specifically the mitigation of insulin resistance. In addition to its impact on glucose metabolism, the treatment also reduced protein breakdown and enhanced muscle anabolic signalling, such as tyrosine kinase activation [57].

The study of Wendler et al. compared two groups of patients with different perioperative nutritional management. One group underwent a 12 h preoperative fast with liquid and solid restriction and resumed a diet 36 h after the surgical intervention; the other underwent a 2 h preoperative fast with a liquid carbohydrate diet and resumed a diet 24 h after surgery. Patients with a longer preoperative fast had higher insulin levels, typical of the insulin resistance condition, in the first 24 h than those subjected to a 2-h fast. In addition, patients submitted to a shortened fast with a carbohydrate-enriched drink and early feeding exhibited lower RGV (residual gastric volume) values, lower insulin resistance, a lower incidence of nausea in the immediate postoperative period, a shorter hospital stay, and better postoperative recovery, compared to the control group [58].

Another systematic review suggests beneficial effects of shortening fasting in patients submitted to cancer surgery, especially in the abdomen area. Preventive administration of carbohydrate-containing liquids, with or without protein, may result in reduced hospital stay, improved glycaemic parameters, changes in the inflammatory profile, and functional capacity [59,60].

The systematic review by Noba and Wakefield showed that preoperative carbohydrate drinks can be safely administered up to 2 h prior to the induction of anaesthesia and do not increase the risk of pulmonary aspiration. Furthermore, they may reduce insulin resistance and alleviate postoperative discomfort, especially in patients undergoing laparoscopic cholecystectomy [50].

The meta-analysis from Ricci and colleagues [61] indicates that PreCL or clear water reduce morbidity and PreCL lowers inflammation, nausea/emesis, and improves carbohydrate homeostasis. Therefore, PreCL can be administered safely in non-diabetic and diabetic patients, and it can be considered a part of ERABS [56].

Nevertheless, other studies did not observe any positive effects of PreCL on post-surgical outcomes, particularly regarding insulin resistance. Moreover, the existing studies often combined control groups that followed different protocols (i.e., fasting or water). Therefore, interpreting the results is not straightforward.

Tong et al. carried out a meta-analysis which compared the effects of different preoperative nutritional approaches on several postoperative outcomes. In particular, these authors considered five preoperative protocols: low-dose oral PreCL, high-dose oral PreCL, intravenous carbohydrates, placebo or water, and fasting. Despite some improved results for the groups undergoing a PreCL protocol, this study did not show any significant difference between the intervention and control groups [32].

## 5. Postoperative Fasting/Early Oral Feeding (POF/EOF)

There is a paucity of data about early refeeding, specifically concerning BS. The nutritional status of patients and extended periods of postoperative fasting (POF) contribute to longer hospital stays following elective gastrointestinal and abdominal wall surgeries. POF is also linked to a higher rate of postoperative complications [62]. Patients who experience POF for a day or more are at an elevated risk for infections, even when other variables are considered. This is particularly prevalent in digestive and colorectal surgeries due to apprehensions about postoperative complications. These apprehensions lead to a longer length of stay (LOS) and higher infection rates, highlighting the need for standardized protocols for enteral or early parenteral nutrition (EPN) [63]. In gastric cancer surgeries, early oral feeding can lead to shorter LOS and improvements in quality of life (QOL) measures, without additional complications. Enteral nutrition has also shown benefits in reducing inflammation and sepsis during the acute phase post-surgery [64]. Major et al. demonstrated that high oral fluid intake on the postoperative day (POD0) is a protective factor. Specifically, for every 100 mL of oral fluid consumed on POD0, the risk of prolonged LOS decreased by 23%, and the readmission risk was lowered by 0.54-fold [65].

The hospital discharge day and postoperative hospital length of stay in the protocol followed by Shimizu et al. were significantly reduced in the total gastrectomy (TG) surgery group, but not in the distal gastrectomy (DG) group. Based on these results, POF appeared to be a likely strategy to shorten the post-operative hospital stay of TG patients without increasing postoperative complications, while in DG patients, given the limited effects on postoperative hospital length of stay and the higher incidence of complications, the authors do not recommend the unselected adoption of EOF, although further confirmatory studies are needed [66].

Lopes et al. observed retrospectively non-infectious general complications in 161 patients undergoing gastrectomy and/or elective esophagectomy and found that EOF can be safely administered. Furthermore, mortality was lower and there was no increased incidence of fistulas in the upper tract compared to enteral feeding. The authors conclude that EOF is safe and feasible for patients undergoing total gastrectomy and/or esophagectomy [67].

In the Sierzega et al. study, EOF was characterized as a clear liquid diet starting on the first postoperative day, with a phased introduction of solid foods on days 2–3. Late oral feeding involved beginning a liquid diet between postoperative days 4 to 6, followed by a gradual switch to solid foods. In a subsequent study by Sierzega et al., focused on a uniform group of Western patients undergoing total gastrectomy for cancer, initiating an early oral diet did not elevate postoperative risks, including damage to the esophago-jugal anastomosis [68].

Bevilacqua et al. also showed that early feeding after BS was independently associated with shorter LOS and suggested that early feeding could explain a significant proportion of the reduction in LOS observed in previous studies of comprehensive ERAS protocols in the bariatric literature [31].

Some studies showed early refeeding is helpful for the restoration of bowel movement [69,70] while postoperative fasting can have negative psychological impact [71]. EOF can be considered a part of ERABS [70].

Table 1 summarizes the main findings of the 20 selected articles, which analysed the various perioperative phases. More specifically, seven analysed preoperative fasting, nine preoperative carbohydrates load, and four postoperative fasting/early refeeding.

## 6. Discussion

This review discusses nutritional treatment in the perisurgical period and specifically the application of ERABS protocols.

Perioperative nutritional care should include avoiding fasting (AF), preoperative carbohydrate loading (PreCL), and early postoperative oral feeding (EOF). ERABS guidelines [22] support AF and PreCL with a low level of evidence, with a weak recommendation for PreCL and strong for AF. EOF presented moderate level of evidence and a strong recommendation. The Italian Consensus Statement supported EOF with level of evidence 1 and strength of recommendation A, but no strong evidence supports PreCL. The results of a Cochrane Review [55] provided very low-quality evidence on the efficacy of PreCL as a part of ERAS protocol.

As for PF, in elective surgery, a 12-h or overnight fasting protocol is commonly followed to minimize gastrointestinal and respiratory complications associated with anaesthesia. However, the literature broadly agrees that shorter fasting periods are beneficial [72].

In addition, some evidence shows that PreCL reduces metabolic complications and loss of lean mass [21,22]. However, regarding PreCL, the data are still controversial, and more well-designed studies are needed to improve the strength of the aforementioned evidence [32].

Regarding POF/EOF, even though there is an important number of studies in the scientific literature attesting to the usefulness, safety, and greater well-being of patients who are fed from the first day after surgery, there are still many reservations in the application of the ERAS nutritional items protocols in general and especially in digestive tract surgery. This is in the hope of avoiding the postoperative complications of anastomotic leaks and postoperative paralytic ileus. In fact, the traditional idea is that feeding of patients after gastric resection should not be started until flatus or defecation have documented the return of bowel function. Conventionally, feeding patients after gastrointestinal surgery has been prescribed only after the return of peristalsis, and today clinicians still debate this topic, despite it having been shown that early feeding can be administered without risk and with potential benefits to patients [72].

In the postoperative period, fasting assumes that oral feeding may not be tolerated during ileus or reduced intestinal motility, and that it may protect the integrity of the newly formed anastomosis, preventing anastomotic leakage. Within 24 h of starvation, metabolic changes including increased insulin resistance and reduced muscle function have been described. This, coupled with malnutrition on admission, and the catabolic effects of surgery, can further compromise the nutritional status of patients and be detrimental to their prognosis [39]. In addition, prolonged fasting has a negative impact on patients’ reported outcomes including severe thirst and dry mouth, emotional fixation on food, and fear over its reintroduction. Anxiety over food reintroduction and increased risk of complications induced by an inadequate nutritional status lend themselves to the hypothesis that patients who undergo excessive fasting prior to surgery may remain fasting longer in the postoperative period. Furthermore, a significant correlation was found between prolonged preoperative fasting and a longer time before shifting to a solid food diet [73].

Rossoni et al. conducted a study on the effectiveness of ERAS protocols in bariatric surgery (BS). The findings indicate that implementing these protocols enhances outcomes like length of stay (LOS) and morbidity/mortality rates in a safe and efficient manner. Barriers to adopting ERABS principles, aimed at expediting patient recovery, often involve inadequate preventive education and guidance for patients, resistance to pre- and postoperative nutritional management, mobilization, and a shift away from narcotic pain relief. A lack of comprehensive prospective research, particularly concerning the nutritional elements, also hampers the efficacy of ERABS. According to ERABS guidelines, post-surgery nutrition should commence promptly, with a liquid diet introduced within the first 24 h after anaesthesia recovery, as tolerated by the patient. Special focus should be given to administering supplements like protein, and chewable vitamins and minerals, notably vitamin B12, iron, and calcium. This study underlines that these supplementations are given in non-uniform timing and not always as soon as possible, disregarding ERABS protocols despite published guidelines [74].

Non-compliance with protocols can also be attributed to logistical and organizational problems.

In one Dutch hospital, the duration of preoperative fasting exceeded by 2.5 times the recommended 6 h for solid foods and 2 h for liquids. Furthermore, one in five patients fasted for more than 18 h for solid foods or 12 h for liquids. One cause could be attributed to a surgery scheduled early in the morning that would prevent breakfast intake, considering the required 6 h of fasting. However, in the same study, a prolonged fast was reported also for surgeries scheduled in the afternoon. Another factor is the patients’ lack of awareness of proper fasting guidelines, despite having been informed in advance, as half of the patients think they should stop eating and drinking before midnight [75].

The European Society of Anaesthesiology recommends that patients drink fluids up to 2 h before surgery [76] to mitigate risks like thirst, hunger, and metabolic changes, including reduced insulin sensitivity, which can lead to severe postoperative complications such as death and infection. Extended fasting could also lead to cardiac stress and hypothermia. Emotional impacts, such as the patient’s sense of losing control due to food and drink restrictions, can also arise. Patients often lack a clear understanding of the need for fasting and the reasoning behind the nil-per-mouth directive [75].

Similarly, the ACERTO project highlighted non-compliance with the ERAS programme with regard to the preoperative fasting time. This finding agrees with that of the BIGFAST multicentre study, previously conducted in 17 Brazilian hospitals which showed that although it is advised to fast the patients for a maximum of 6 h, half of them do not eat for more than 12 h before surgery. One possible explanation for this discrepancy is that preoperative fasting has not yet been given due attention and perhaps an appropriate protocol for patient orientation has not yet been created [77].

Also, an Australian study underlines the difficulty for some patients to understand the information about their care, which in turn could negatively impact the overall postoperative outcomes. The conclusions of this study encourage health practitioners to adopt a more clear and simple language to improve the decision-making process and achieve shared decisions for the best outcomes [78].

Effective management of perioperative fasting requires both the training of doctors and nurses and the cooperation of patients. A compliance rate > 70% is usually considered satisfactory. However, the median preoperative fasting time for liquids is still longer than that recommended, due to the uncertainty linked to delays in scheduled times. Moreover, the study suggests reducing preoperative fasting for solid foods, still prohibited before surgery, to enhance the comfort and the recovery [79].

Small differences can be detected between various consensus documents.

Canadian ERABS guidelines established 2 h of fasting from clear fluids and 6 from solid foods, but disclaim evidence for the necessity of PreCL in BS. Furthermore, they do not recommend unequivocally resuming feeding on POD0 [80].

These statements are consistent with the Italian ones, except for POF which is discouraged to decrease postoperative complication risks, which include infections, delayed healing and restoration of bowel movements, together with a worse psychological relationship with food (emotional fixation on food, and a phobia about the reintroduction of food). Furthermore, early refeeding can reduce mortality and LOS [28]. In summary, following the ERABS protocol, the Italian Consensus Statement underlines a firm grade A recommendation for avoiding preoperative fasting. It advises that clear liquids and solid food are recommended up to 2 h and 6 h, respectively, before the induction of anaesthesia. Moreover, early refeeding, meaning the early postoperative resumption of oral feeding, is recommended with level 1 of evidence. Instead, no strong evidence supports preoperative oral carbohydrate loading in bariatric surgery.

### Gaps in the Research

We have identified some gaps in the research about practices that could be added to ERABS protocols; nevertheless, rigorous trials are missing.

One gap in the research concerns the use of specialized pro-resolving lipid mediators (SPM) for acute inflammation linked to surgical procedures and for chronic inflammation related to obesity [81,82]. Recent findings indicate that the metabolism and levels of these compounds can influence surgical outcomes, including weight loss and other health aspects (i.e., cognitive decline) [83].

Patients with obesity have a reduced SPM production [84]. Some evidence shows SPM can reduce pain, infections, and the need for drugs, and can enhance recovery after surgical procedures, and it has been suggested that SPM can be administered as nutritional supplement [82,85,86,87]. Could this practice improve recovery?

BS is associated with changes in the gut microbiota (GM) that influence outcomes, acting on the neuroendocrine axis and absorption of nutrients [88]. Mounting evidence shows an appropriate GM is linked to metabolic health and healthy aging while loss of species reduces quality of life and lifespan [89,90]. Can the handling of microbiota via prebiotics, probiotics and symbiotics provide benefits in pre- and postoperative phases, even taking into account that antibiotic treatments are mandatory for BS [91,92,93,94]? A recent systematic review shows that probiotics protect from loss of diversity and changes in gut microbial composition after intake of antibiotics [95] and American joint guidelines from different Scientific Societies support their role in cases of small intestinal bacterial overgrowth after BS [18].

Can the supplementation with essential amino acids in early refeeding/first days after intervention enhance recovery and reduce complications? This type of supplementation has shown efficacy in many types of surgery, counteracting malnutrition and promoting healing [96,97,98,99,100,101,102,103,104,105].

Finally, ERABS protocols do not recommend body composition analysis to evaluate the nutritional status of the patient, although ESPEN guidelines suggest a bioelectrical impedance analysis (BIA) or other enhanced technique, before surgery [106].

The validity of BIA in severe obesity is still debated, and it would need specific new equations [107,108,109].

## 7. Strengths and Limitations

The main strength of this review is its consideration of protocols based on guidelines from the ERABS Society and other major Scientific Societies. Furthermore, we highlighted the lack of proper application of the ERABS protocol and clarified the reasons why it should be routinely used in clinical practice.

However, due to the limited application of this protocol there are still few well-designed studies. Consequently, one limitation of this review is the inclusion of observational studies alongside intervention ones.

## 8. Conclusions

ERABS protocols must be implemented and supported to reduce complications, stress, and anxiety and promote healing after BS.

Some practitioners do not comply with the principles of ERABS protocols, in particular the elimination of preoperative fasting, for reasons that are not evidence-based.

Some interesting complementary practices like the use of certain supplements require further research before they are applied.

## Figures and Tables

**Figure 1 ijerph-20-06899-f001:**
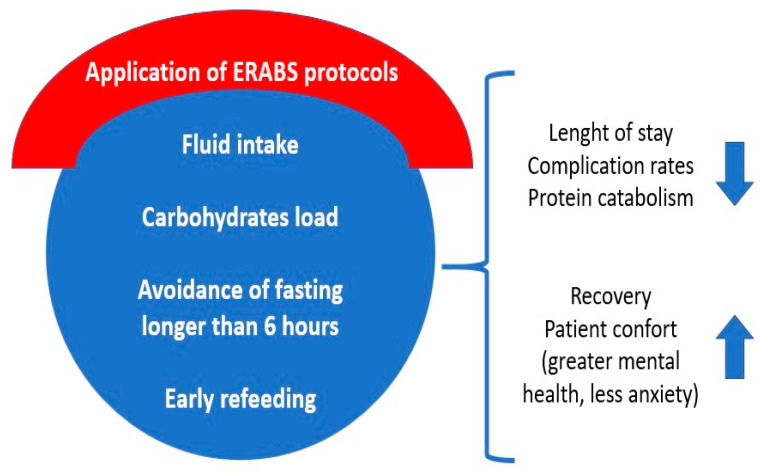
Summary of ERABS nutritional items and their effects.

**Figure 2 ijerph-20-06899-f002:**
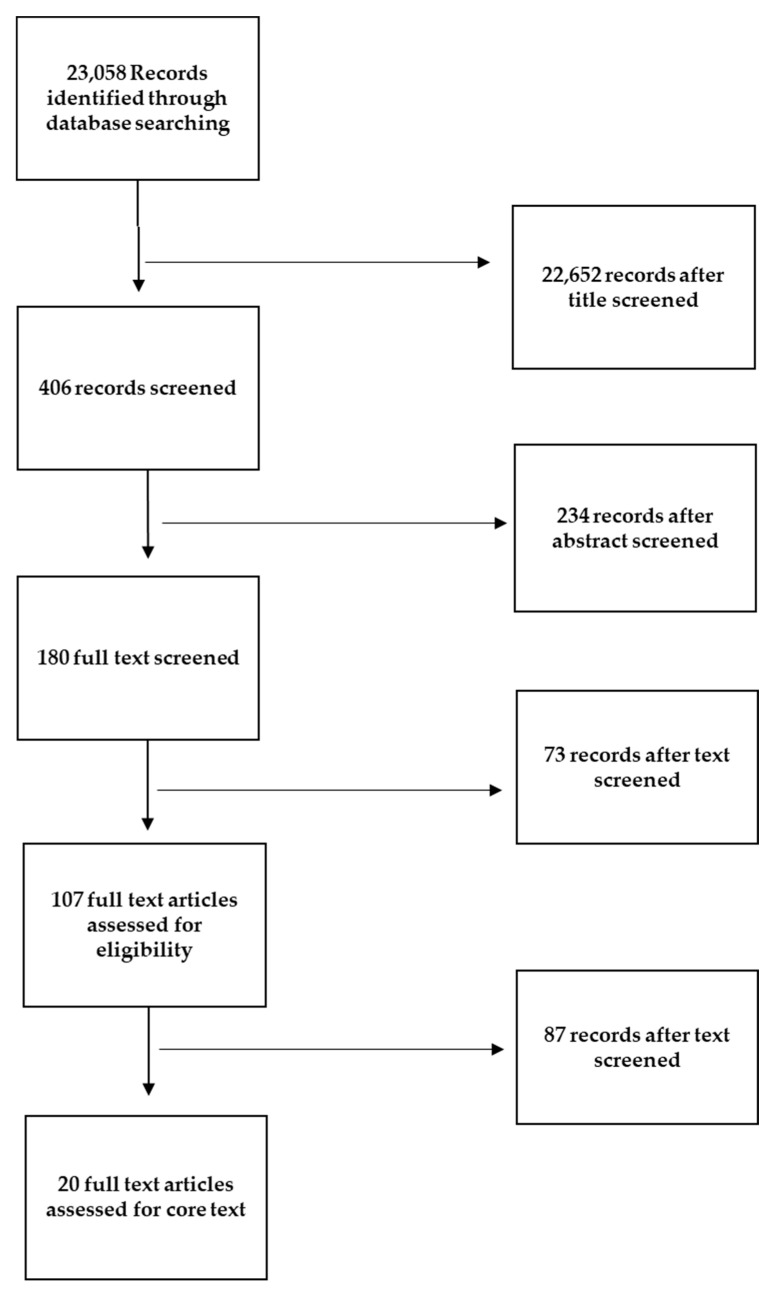
Flow diagram illustrating the process of our review, screening, and article selection.

**Table 1 ijerph-20-06899-t001:** Summary of characteristics of included studies.

ERABS Item	Publication	Type	Main Findings
	[35]	Clinical guidelines	Avoiding PF is a part of ERAS
	[39]	article	Quit PF
Preoperative fasting	[40]	Systematic review	Many anaesthetists follow outdated practices
	[43]	Retrospective analysis	Need for aspiration is uncommon
	[45]	Retrospective study	Shorter PF favours recovery
	[51]	Comment	PF can be reasonable in some circumstances
	[65]	Observational study	Fluid intake reduces hospital stay
	[32]	Systematic review	Evidence about PreCL is low
	[37]	review	PreCL is beneficial
	[38]	review	PreCL reduces discomfort
Preoperative carbohydrates load	[52]	RCT	PreCL does not reduce nausea
	[54]	Review	PreCL reduce IR and improve SR
	[55]	Systematic review	PreCL reduces LOS
	[56]	RCT	PreCL does not rise risks
	[57]	Meta-analysis	PreCL can reduce LOS in major surgery
	[58]	RCT	PreCL accelerates recovery after RY
Postoperative fasting/early refeeding	[31]	Retrospective study	Early feeding shortens LOS
	[49]	Systematic review	Post-operative fasting has negative effects
	[69]	Retrospective study	Early feeding improves food tolerance
	[70]	Retrospective study	Early fluid intake reduces pain and complications

Summary of the main findings of the 20 articles included in this review, relevant to the analysis of the various perioperative phases, including seven for preoperative fasting (PF), nine for preoperative carbohydrates load (PreCL), and four for postoperative fasting (POF)/early refeeding. Length of stay (LOS), Roux-en-Y (RY).

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
