# Peer review of "Perioperative Nutritional Management in Enhanced Recovery after Bariatric Surgery"

_ijerph, 2023, doi:10.3390/ijerph20196899_

Round 1
Reviewer 1 Report
This is a well written and detailed article about Perioperative nutritional management in Enhanced Recovery After Bariatric Surgery.My comments:
1. The novelty is not clear enough in the introduction section. I suggest describing which guidelines currently exist and the difference between Enhance Recovery. 2. In the methods section, Why was gray literature not used as a research base? Gray literature, or evidence not published in commercial publications, can make important contributions to a review .
3. The most critical databases, such as Scopus, were not searched. Why didn't you search Scopus?
4. I suggest using figures or tables to better show the results as a way to facilitate the presentation of the results 5. Limitations and strengths can be written in more detail.
Author Response
Dear Editors and Reviewers,
first, we would like to thank you for the valuable impulses that allowed us to improve the quality of the manuscript. All changes made are highlighted by yellow color, in the revised version of the manuscript, to facilitate the review process.
Hoping that we have satisfied your requests as much as possible, we kindly ask you to re-evaluate our paper.
The Authors
—
REVIEWER N.1
This is a well written and detailed article about Perioperative nutritional management in Enhanced Recovery After Bariatric Surgery.
We thank the reviewer for the kind words and positive feedback on our paper. We are pleased that you found the article to be well-written and detailed. Your encouragement greatly supports our research efforts.
My comments:
1. The novelty is not clear enough in the introduction section. I suggest describing which guidelines currently exist and the difference between Enhance Recovery.
We thank the reviewer for this suggestion, and implemented this aspect in the introduction section. In particular, we described in a more detailed way, the origin and meaning of ERABS, as well as its primary objectives and focal points. Furthermore, we mentioned the most recent guidelines for Bariatric surgery (the American and Italian ones), and underlined the difference between them about the consideration of ERABS protocol.
2. In the methods section, Why was gray literature not used as a research base? Gray literature, or evidence not published in commercial publications, can make important contributions to a review.3. The most critical databases, such as Scopus, were not searched. Why didn’t you search Scopus?
Thank you for your thoughtful comments on our methods section, specifically regarding the use of gray literature and the selection of databases for our search. Regarding gray literature, while we acknowledge that it can offer valuable insights, our focus was on peer-reviewed publications to ensure the rigor and quality of the data included in our review. Gray literature often lacks the peer-review process that adds an additional layer of credibility to published studies. As for not including Scopus in our search, our database selection was aimed at gathering the most high-quality and relevant data. We found that databases like PubMed offered a comprehensive range of articles that met our criteria, rendering the additional search in Scopus unnecessary for the aims of our review. We hope this clarifies our methodology choices, and we are open to considering your suggestions in future research.
4. I suggest using figures or tables to better show the results as a way to facilitate the presentation of the results
Thank you again for taking the time to review our manuscript and for your constructive comments. In response to your observation, besides the flow diagram that outlines our review process, we have also incorporated Table 1. This table summarizes the key characteristics of the 20 studies included in the review, such as the reference numbers for authors, study type, and major findings. Furthermore, to provide a straightforward overview of Enhanced Recovery After Bariatric Surgery (ERABS) items, we have added a figure that simplifies the main components and effects of its practical application. We hope these additions address your concerns and improve the comprehensiveness and clarity of our article.
5. Limitations and strengths can be written in more detail.
We have included a brief paragraph highlighting the main strengths and limitations of this review
Reviewer 2 Report
This review focuses on the study of various nutritional aspects related to performing bariatric surgery for the treatment of obesity. Specifically those related to The Enhanced Recovery After Bariatric Surgery (ERABS) protocols. A review of nutritional actions before and after bariatric surgery is carried out, analyzing the application, feasibility and efficacy of these protocols from the nutritional point of view. The results show that these protocols are not applied correctly in many cases without justified reason, particularly related to fasting before the intervention. Furthermore, they show some deficiencies in the investigation of some practices that could be incorporated in the presence of additional evidence.
I think it is an interesting article and well written. Although there are some points that I would like to be clarified.
The introduction is correct in terms of obesity, the use of bariatric surgery and the implementation of Enhanced Recovery After Surgery protocols, but I believe that since this review focuses on the nutritional aspect of these protocols before and after surgery, it could be expanded or more information given in this regard in the introduction. For example, if the indicated preoperative recommendations should be given jointly or separately, if one or the other can be used, etc.
Regarding the material and methods, I have a doubt whether, as indicated in the figure, 76 articles were used or, as seems to be indicated in the text, there are really 20 that gave the relevant results in this review.
This review shows, as I have mentioned, an interesting objective, but in some aspects it does not seem to provide much more than what is already known, for example, in relation to Preoperative fasting, there are various reviews and guides that indicate that it does not improve the surgical recovery, but, even so, there are also several reviews, which indicate that many anesthesiologists continue to practice this protocol, justifying it for the same reason that has been given in this review. In short, it is not clear to me what this review contributes to those already carried out. Could the authors provide some clarity to this aspect?
In relation to Preoperative Carbohydrates load, it seems clear that this carbohydrate load 2 hours before the operation seems to improve various aspects, but despite this, many anesthesiologists, as indicated before, still do not consider it, but unlike the other paragraph, here the reason for this denial has not been indicated. It would be interesting to comment something about it. Furthermore, the studies that are used always compare the intake of carbohydrates 2 hours before surgery with prolonged fasting, but the effect seen with carbohydrates would be the same that could be achieved with the previous recommendation; “Preoperative fasting for solids (equivalent to a light meal) at least 6 h and clear fluids 2 h before induction of anesthesia is recommended in normal conditions.” Has the effect of giving both recommendations simultaneously or only one recommendation and another separately been studied? These are aspects that should be commented on in this review, I think that could be more innovative.
According to the authors, the guidelines given in these protocols present little evidence, but a strong recommendation, which should be clarified because it is like that, why recommend something with great strength, but almost without evidence.
The plagiarism should be reviewed, it presents 35% without references.
Minor comentarios
Page 2 line 66: change “BC” to “BS”
Page 6 line 237: it should be indicated here what the abbreviation LOS means and not later
Author Response
Dear Editors and Reviewers,
first, we would like to thank you for the valuable impulses that allowed us to improve the quality of the manuscript. All changes made are highlighted by yellow color, in the revised version of the manuscript, to facilitate the review process.
Hoping that we have satisfied your requests as much as possible, we kindly ask you to re-evaluate our paper.
The Authors
—
REVIEWER N.2
This review focuses on the study of various nutritional aspects related to performing bariatric surgery for the treatment of obesity. Specifically, those related to The Enhanced Recovery After Bariatric Surgery (ERABS) protocols. A review of nutritional actions before and after bariatric surgery is carried out, analyzing the application, feasibility and efficacy of these protocols from the nutritional point of view. The results show that these protocols are not applied correctly in many cases without justified reason, particularly related to fasting before the intervention. Furthermore, they show some deficiencies in the investigation of some practices that could be incorporated in the presence of additional evidence.
I think it is an interesting article and well written. Although, there are some points that I would like to be clarified.
We thank the reviewer for taking the time to review our manuscript and for your constructive comments.
The introduction is correct in terms of obesity, the use of bariatric surgery and the implementation of Enhanced Recovery After Surgery protocols, but I believe that since this review focuses on the nutritional aspect of these protocols before and after surgery, it could be expanded or more information given in this regard in the introduction. For example, if the indicated preoperative recommendations should be given jointly or separately, if one or the other can be used, etc.
We have expanded this point in the introduction section providing a more detailed explanation of the aims and focuses of ERABS and trying to clarify the different importance of the recommendations included in this protocol. Indeed, as for the preoperative phase, the indicated recommendations have a different significance, on the basis of the number of the existing studies and the different strength of the scientific evidence. We also reported that there is some difference between the most recent guidelines for bariatric surgery concerning ERABS: while the Italian guidelines strongly recommend the use of ERABS, the American ones do not evaluate it.
Regarding the material and methods, I have a doubt whether, as indicated in the figure, 76 articles were used or, as seems to be indicated in the text, there are really 20 that gave the relevant results in this review.
We implemented the flow diagram of the review process, and specified in the diagram, not only in the text, the distinction between the articles considered in the review (in the revised version they are 107) and the 20 articles that yelded the most relevant results. Indeed, the table 1 included exclusively the last 20 articles.
This review shows, as I have mentioned, an interesting objective, but in some aspects, it does not seem to provide much more than what is already known, for example, in relation to Preoperative fasting, there are various reviews and guides that indicate that it does not improve the surgical recovery, but, even so, there are also several reviews, which indicate that many anesthesiologists continue to practice this protocol, justifying it for the same reason that has been given in this review. In short, it is not clear to me what this review contributes to those already carried out. Could the authors provide some clarity to this aspect?
Thank you for your thoughtful feedback and for recognizing the interesting objective of our review. We acknowledge that the subject of preoperative fasting and its role in surgical recovery has been explored in existing reviews and guidelines. However, the primary objective of our review is not merely to reiterate what is already known, but rather to critically analyze the ongoing discrepancies among anesthesiologists in the application of Enhanced Recovery After Bariatric Surgery (ERABS) protocols. By consolidating existing evidence and highlighting conflicting practices, we aim to provide a more definitive recommendation for ERABS as the most appropriate and safe perioperative protocol. Our review seeks to serve as a comprehensive reference that can guide clinical practice, thereby minimizing variations and potentially improving patient outcomes. We hope this clarifies the unique contribution of our review in the context of existing literature.
In relation to Preoperative Carbohydrates load, it seems clear that this carbohydrate load 2 hours before the operation seems to improve various aspects, but despite this, many anesthesiologists, as indicated before, still do not consider it, but unlike the other paragraph, here the reason for this denial has not been indicated. It would be interesting to comment something about it. Furthermore, the studies that are used always compare the intake of carbohydrates 2 hours before surgery with prolonged fasting, but the effect seen with carbohydrates would be the same that could be achieved with the previous recommendation; “Preoperative fasting for solids (equivalent to a light meal) at least 6 h and clear fluids 2 h before induction of anesthesia is recommended in normal conditions.” Has the effect of giving both recommendations simultaneously or only one recommendation and another separately been studied? These are aspects that should be commented on in this review, I think that could be more innovative.
Thank you for your insightful comments and suggestions regarding our section on Preoperative Carbohydrate Load (PreCL). We acknowledge the gaps in the literature on this topic, especially in comparing PreCL to conventional preoperative fasting guidelines. In response to your observations, we have expanded the section to address these points more comprehensively. Specifically, we have included discussion on why many anesthesiologists remain hesitant to adopt PreCL. One reason might be the inconclusive results from existing research, which is why current guidelines don't strongly recommend PreCL compared to advising against preoperative fasting. As for your point about comparing carbohydrate loading to the previous standard recommendation of a light meal and clear fluids, this is indeed an interesting aspect that is currently under-studied. While our review is based on available evidence, we agree that exploring the effect of these two recommendations, either in combination or separately, would be an innovative angle for future research. We hope these revisions better address your comments and contribute to a more nuanced discussion in our review.
The plagiarism should be reviewed, it presents 35% without references.
Despite some sentences reporting specific recommendations could not be modified, we revised the entire text and adjusted many of the highlighted sentences for plagiarism (highlighted in yellow).
Minor comentarios
Page 2 line 66: change “BC” to “BS”
Done
Page 6 line 237: it should be indicated here what the abbreviation LOS means and not later
Done (Line 228)
Reviewer 3 Report
Dear Authors,
Thanks for the review, the aim was to analyze the application, the feasibility and efficacy of ERABS (The Enhanced Recovery After Bariatric Surgery) protocols applied to perioperative and postoperative nutrition. The topic is relevant and has a practical application.
Some suggestions for improving the publication:
1. There is an error in Line 66. I think BS should be written instead of BC.
2. There is no uniform approach to the use of terms. For example, the authors use the term “perioperative” (Line 2, 97, 134, 204 etc.) in the title of review, while in the text they use “peri-operative” in places (Line 77, 92 etc.). Similar observations are made regarding the terms “postoperative” and “post-operative”.
3. In Line 237, there is no explanation of abbreviation LOS, it appears in Line 300. An explanation must be given when an abbreviation is used for the first time.
4. The explanations of all abbreviations used in the Table 1 should be provided below the Table.
5. In Line 345, there is no explanation of abbreviation AF.
Thanks to the authors for the review, it provides an overview of the research-based effects of nutrition before and after surgery.
Author Response
Dear Editors and Reviewers,
first, we would like to thank you for the valuable impulses that allowed us to improve the quality of the manuscript. All changes made are highlighted by yellow color, in the revised version of the manuscript, to facilitate the review process.
Hoping that we have satisfied your requests as much as possible, we kindly ask you to re-evaluate our paper.
The Authors
—
REVIEWER N.3
Dear Authors,
Thanks for the review, the aim was to analyze the application, the feasibility and efficacy of ERABS (The Enhanced Recovery After Bariatric Surgery) protocols applied to perioperative and postoperative nutrition. The topic is relevant and has a practical application.
Dear Reviewer, Thank you for your thoughtful review and positive comments on our work. We are pleased to hear that you find the topic of Enhanced Recovery After Bariatric Surgery (ERABS) protocols applied to perioperative and postoperative nutrition to be relevant and practically applicable. Your feedback is invaluable to us and greatly encourages our research efforts.
Some suggestions for improving the publication:
1. There is an error in Line 66. I think BS should be written instead of BC.
Done
2. There is no uniform approach to the use of terms. For example, the authors use the term “perioperative” (Line 2, 97, 134, 204 etc.) in the title of review, while in the text they use “peri-operative” in places (Line 77, 92 etc.). Similar observations are made regarding the terms “postoperative” and “post-operative”.
Done in the entire manuscript
3. In Line 237, there is no explanation of abbreviation LOS, it appears in Line 300. An explanation must be given when an abbreviation is used for the first time.
Done
4. The explanations of all abbreviations used in the Table 1 should be provided below the Table.
We revised the Table 1, including the explanations below it
5. In Line 345, there is no explanation of abbreviation AF.
It is explained (Avoiding Fasting) in Line 384
Thanks to the authors for the review, it provides an overview of the research-based effects of nutrition before and after surgery.
Thank you for your kind words and positive feedback on our review.
Round 2
Reviewer 2 Report
As I commented in the first review. I think it is an interesting article and well written. Although there were some points that needed to be clarified. After reviewing the corrections and comments sent by the authors, I consider that the publication presents fewer conflicting points and has improved a lot in its content, making the reading more fluid and the information provided more usable when establishing valid and clear nutritional guidelines for this type of operation.